# Who Is Afraid of Romantic Relationships? Relationship Fears and Their Connection with Personal Values and Socio-Demographic Variables

**DOI:** 10.3390/bs15020191

**Published:** 2025-02-11

**Authors:** Eugene Tartakovsky

**Affiliations:** The School of Social Work, Tel Aviv University, Tel Aviv-Yafo 69978, Israel; evgenyt@tauex.tau.ac.il

**Keywords:** fear of romantic relationships, emerging adults, Jewish and Palestinian Israelis, personal values, gender differences

## Abstract

This study investigates the fears of romantic relationships. Based on Schwartz’s theory of values, we built a comprehensive inventory of the fears that young people seeking romantic relationships may experience. We tested the fears’ structure and the connections with personal values and socio-demographic variables. The study was conducted in Israel using a community sample of young Jews and Palestinians without romantic partners (n = 1083, 57% female, age 18–30). We discovered ten basic fears clustered into three groups (concerns). *The ineptitude concern* combines two fears: failing one’s partner’s expectations and failing expectations of one’s relatives and friends. *The subjugation concern* combines four fears: loss of independence, boredom, sexual frustration, and thwarting one’s achievements. Finally, *the abuse concern* combines four fears: losing control over one’s resources, being hurt physically or sexually, harming relationships with one’s relatives and friends, and being accused of inappropriate behavior. The ranking of concerns was identical among men and women, with ineptitude being the strongest concern, followed by subjugation and abuse. The ineptitude concern was associated with a higher preference for self-transcendence vs. self-enhancement values. The subjugation concern was associated with higher preferences for openness to change vs. conservation and self-enhancement vs. self-transcendence values. The abuse concern was associated with a higher preference for conservation vs. openness to change values. The ineptitude concern was stronger among younger ages, females, and Jews. The subjugation concern was stronger among older ages, males, and less religious people. Finally, the abuse concern was stronger among younger ages, males, religious people, and Palestinian Israelis. Socio-demographic variables affected relationship concerns directly and indirectly through their connection with personal values. The present study advances the theory of values connecting context-specific and general motivations. The results obtained will be helpful in youth counseling to promote satisfactory decisions regarding romantic relationships.

“If you are afraid of loneliness, do not marry.” Anthon Chekhov

## 1. Introduction

Romantic relationships are interpersonal interactions based on emotional and physical attraction that could lead to a long-term intimate connection ([3]). Romantic relationships fulfill the basic human needs of being loved, feeling safe, and bonding with a significant other. Therefore, they are paramount for individual well-being ([1]). However, many adolescents and young adults do not have and do not want a romantic partner, and a growing number of people worldwide remain single throughout their lives ([7]; [19]; [27]). Factors averting people from romantic relationships have been scarcely investigated. In this study, we are trying to understand what people fear in romantic relationships.

The study aims to investigate the motivational foundations of relationship fears and, at a more abstract theoretical level, to reveal the connections between context-specific and general motivations. The study is based on the theory of values ([21]), which considers personal values as cognitive constructs reflecting general motivational goals and directing people’s cognition and behavior ([20]). The study consisted of three stages. First, we created a relationship fears inventory using scientific literature and personal interviews. After that, we tested the inventory’s structure using exploratory and confirmatory analyses. Finally, we investigated the connections between relationship fears, personal value preferences, and socio-demographic variables. We conducted a study in Israel using a large community sample of young Jews and Palestinians who had no romantic partners.

### 1.1. Studies on Relationship Fears

There are several lines of research on relationship fears. Some researchers focus on worries related to marriage. Interviewing different socio-demographic groups, they created a list of marriage worries. These worries include the disturbing of work-family balance, conflicts, abuse, loss of freedom, problems with decision-making, infidelity, control over finances, not being a good spouse, sex problems, and diminished contact with relatives and friends ([8]; [10]; [11]).

Another line of research focuses on the advantages of being single. Researchers assume that being single permits an individual to achieve important goals such as advancing one’s career, multiplying mating, having more time for oneself, and spending more time with one’s relatives ([2]). People may avoid romantic relationships because they threaten the goals they may reach by being single.

Finally, the third line of research investigates dating anxieties, i.e., fears related to the meeting with a prospective romantic partner. Dating obviously differs from marriage; however, the lists of fears/anxieties within the two lines of research are similar, with the only difference being that dating anxieties are more situation-specific and transitional than marriage fears ([1]; [12]).

Researchers found that relationship fears vary depending on socio-demographic characteristics. For instance, romantic relationships threaten young people and men more because freedom, career, and multiple mates are more important to these groups. On the other hand, saving resources by being single is more important for older people, because they have accumulated more resources ([2]). Studies focusing on dating anxiety found stronger distress among men and younger participants. The researchers related these findings to the lesser experience of young people and stronger social pressure on men to be more proactive in relationships ([12]).

The main limitation of previous studies on relationship fears is that most have not been theory-based. Usually, they list the advantages of being single, the disadvantages of being in relationships, or dating anxieties, as reported by participants. Some researchers grouped the scale items; however, they did not investigate connections between the groups and had no theoretical basis for formulating hypotheses about the fears’ structure (e.g., [2]). We based our study of relationship fears on the theory of values, a general motivational theory ([21]), which permitted us to investigate the structure of relationship fears and their connections with general motivational goals and socio-demographic variables.

### 1.2. Theory of Values

The theory of values ([21]) defines values as desirable trans-situational goals guiding people’s lives. Thus, personal value preferences reflect the individuals’ general motivational goals, which affect the perception of reality and direct behavior. The theory specifies a comprehensive set of twelve motivationally distinct values, some further divided into lower-level values ([22]). The theory includes the following values: power (dominance and resources), achievement, hedonism, stimulation, self-direction (thought and action), universalism (nature, concern, and tolerance), benevolence (caring and dependability), humility, conformity (rules and interpersonal), tradition, security (personal and societal), and face.

The theory assumes the existence of dynamic relations between values: the pursuit of each value has consequences that may conflict or may be congruent with the pursuit of other values. The conflicts and congruities among values yield an integrated structure of four higher-order values arrayed along two axes. Openness to change values (including self-direction and stimulation) emphasize readiness for new ideas, actions, and experiences. They contrast with conservation values (including conformity, tradition, and security) that emphasize self-restriction, order, and preserving the status quo. Self-enhancement values (including power and achievement) emphasize pursuing personal interests. They contrast with self-transcendence values (including universalism and benevolence) that emphasize transcending personal interests for the sake of others. Finally, the theory assumes that self-transcendence and openness to change values express the goals of growth and self-actualization and are more likely to motivate people when they are free of anxiety. The self-enhancement and conservation values are directed toward protecting the self against anxiety and threats.

Group differences in values have been investigated in several studies. Gender differences in most values have been consistent across cultures, with men reporting higher preferences for self-enhancement and openness to change and women—for self-transcendence values; gender differences in conservation values were less consistent ([23]). Age was positively correlated with conservation and negatively correlated with openness to change values; education was positively correlated with openness to change values; and religiosity was positively correlated with conservation values ([22]). Finally, Israeli studies found that Palestinian Israelis, compared to Jewish Israelis, have higher preferences for conservation vs. openness to change and self-enhancement vs. self-transcendence values ([25], [26]).

Following Schwartz’s theory of values, we assumed that relationship fears express context-specific motivations. We further assumed that people avoid romantic relationships because they fear that such relationships may prevent them from attaining the general motivational goals expressed in their value preferences. Thus, we connected each relationship fear to a specific value, assuming that people with a high preference for a specific value experience a high level of relationship fear associated with this value. Finally, we assumed that the group differences in value preferences may explain group differences in relationship fears.

### 1.3. Study Hypotheses


**1.** **The structure of relationship fears.** Relationship fears form a circumplex paralleling the values’ circumplex. Similarly to values, the fears’ circumplex may be partitioned into four clusters (relationship concerns), each associated with a higher-order value:1.1The first cluster, called *the subjugation concern*, includes the fears of losing independence, being bored, and being sexually frustrated.1.2The second cluster, called *the depletion concern*, includes the fears of thwarting one’s achievements and losing control over one’s resources.1.3The third cluster, called *the abuse concern*, includes the following relationship fears: being endangered physically and sexually, being accused of inappropriate behavior, and harming relationships with relatives and friends.1.4The fourth cluster, called *the ineptitude concern*, includes fears of failing a partner’s expectations and failing to meet the expectations of one’s relatives and friends.1.5Relationship concerns derived from opposing poles of values axes (the subjugation and abuse and the depletion and ineptitude concerns) contradict each other and are located on opposite sides of the circumplex.**2.** 
**Connections between values and relationship concerns:**
2.1The subjugation concern is associated with a higher preference for openness to change vs. conservation values, and the abuse concern is associated with a higher preference for conservation vs. openness to change values.2.2The depletion concern is associated with a higher preference for self-enhancement vs. self-transcendence values, and the ineptitude concern is associated with a higher preference for self-transcendence vs. self-enhancement values.**3.** 
**Connections between socio-demographic variables and relationship concerns:**
3.1Younger people have stronger inaptitude and abuse concerns.3.2Men have stronger subjugation and depletion concerns, and women have stronger abuse concerns.**4.** 
**Connections between socio-demographic variables and values:**
4.1Being a man is associated with higher preferences for openness to change vs. conservation and self-enhancement vs. self-transcendence values.4.2Age is associated with a higher preference for conservation vs. openness to change values.4.3Education is associated with a higher preference for openness to change vs. conservation values.4.4Religiosity is associated with a higher preference for conservation vs. openness to change values.4.5Compared to Jewish Israelis, Palestinian Israelis have a higher preference for conservation vs. openness to change and self-enhancement vs. self-transcendence values.


## 2. Methods

### 2.1. Sample

This study used a community convenience sample of 1083 participants. Adults aged 18–30 who did not have a romantic partner were invited to participate in the study. The average age was 25.1 (*SD* = 3.32, range 18–30). 57% were females. The average education was 13.6 years (*SD* = 2.00, range 11–21). 72% of the participants were Jewish, 20% were Muslims, 4% were Christians, 2% were Druses, and 2% were others. 14% of the participants were atheists, 51% were secular, 24% were traditional, and 11% were religious. Immigrants constituted 5% of the sample. The following groups were overrepresented in the sample: women, atheists and secular, highly educated, Israeli-born, and Muslims.

### 2.2. Procedure

The Tel Aviv University Review Board approved the study. Undergraduate students who participated in a senior research seminar (a third-year BA course) distributed the questionnaires as a course assignment. Students participating in the seminar lived in different regions of the country, ensuring a geographically heterogeneous sample. The anonymity of the participants was ensured, and all participants signed an informed consent form. The questionnaires were distributed using Google Forms through WhatsApp, Facebook, and e-mail. The study was conducted in Hebrew. The participants did not receive compensation.

### 2.3. Instruments

**Relationship fears.** We built an inventory measuring relationship fears in several steps. First, we gathered the advantages of staying single, marriage fears, and dating anxieties mentioned in the literature and reformulated them as relationship fears. In addition, we conducted interviews with about 50 young people from different ethno-religious groups in Israel, asking them about their fears related to romantic relationships. Thus, we created a comprehensive list of relationship fears. After that, with a group of students applying the inter-judges’ agreement, we discarded repeated items and reformulated some items to make them clearer. Then, we used the inter-judges’ agreements to assign the items to specific relationship fears. Thus, we created ten scales measuring fears people experience when considering romantic relationships.

The scale measuring relationship fears was called the Relationship Fears Inventory (RFI). The scale comprised 46 items allocated into ten relationship fears (Table A1 in Appendix A). Each fear was measured by 3–8 items. The participants were asked to what extent they feared the negative consequences when considering romantic relationships. They answered on a 6-point scale, from 1—*not at all* to 6—*very much*. Example items: “I will not be good enough to care for my girl/boyfriend.” “I will lose my independence.” “I will lose control over my financial resources.” The internal consistency of all scales measuring relationship fears was high (Cronbach alphas 0.75–0.94). Scale scores were calculated as means of the corresponding items. Higher scores indicate stronger relationship fears.

**Personal value preferences.** Personal value preferences were measured using the Portrait Values Questionnaire, PVQ-R ([22]). This questionnaire consists of 57 items. Each item portrays an abstract person describing their goals, aspirations, and wishes that indicate the importance of a specific value. Respondents replied how similar the described person is to them on a 6-point scale, from 1 (*not like me at all*) to 6 (*very much like me*). Item example (Conformity): “It is important to him/her to avoid upsetting other people.” Cronbach’s alphas of the four higher-order values were 0.83–0.89. The higher-order values on the axes’ poles were strongly negatively correlated: *r* = −0.74 for openness to change—conservation and *r* = −0.59 for self-transcendence—self-enhancement. To avoid the multicollinearity problem, we used axes’ scores, built by subtracting the scores of one pole of an axis from the other. 

To correct for individual differences in using circumplex scales, each participant’s responses were centered on their means for the values and fears scales used in the present study. To do so, the mean of all items included in the scale was subtracted from subscale scores. Thus, the mean of 57 value scale items was subtracted from the value axes scores, and the mean of 46 relationship fear scale items was subtracted from the scores of 10 fear scales.

**Socio-demographic variables.** Five socio-demographic variables were included in the analysis: age, gender (1—*man*, 2—*woman*), education (number of years studied), ethnicity (1—*Palestinian Israeli*, 2—*Jewish Israeli*), and religiosity (1—*atheist*, 2—*secular*, 3—*traditional*, and 4—*religious*).

### 2.4. Statistical Analyses

We tested the structure of relationship fears in two steps. First, we calculated the mean-centered scores for the ten relationship fears using the appropriate items and applied multidimensional scaling (MDS) to the ten fears. We used the Multidimensional Scale module in SPSS v.30.0.0 to conduct MDS. After establishing the inventory’s structure in MDS, we retested it using a Confirmatory Factor Analysis (CFA) of the ten relationship fears. After that, we used Structural Equation Modeling (SEM) to test the connections between socio-demographic variables, values, and relationship fears. We conducted CFA and SEM using Mplus v.8.11.

## 3. Results

### 3.1. Structure of Relationship Fears

**Multidimensional scaling.** Figure 1 presents a two-dimensional MDS configuration of the Relationship Fears Inventory (RFI). The goodness of fit indexes demonstrated a good fit: Stress = 0.070; RSQ = 0.971. The existence of three separate clusters (ineptitude, subjugation, and abuse) was confirmed. However, the two fears (thwarted achievements and losing control over one’s resources) hypothesized to form the fourth cluster (the depletion concern) were separated between the subjugation and abuse concerns. Thus, the relationship fears formed three clusters (relationship concerns). The first cluster, called *the ineptitude concern*, combined two fears: failing relatives and friends’ expectations and failing a partner’s expectations. The second cluster, called *the subjugation concern*, combined four fears: loss of independence, boredom, sexual frustration, and thwarted achievements. Finally, the third cluster, called *the abuse concern*, combined four fears: losing control over one’s resources, being endangered physically or sexually, harming relationships with one’s relatives and friends, and being accused of inappropriate behavior. The three concerns were negatively correlated with each other (men/women): subjugation with ineptitude (*r* = −0.44/−0.33), subjugation with abuse (−0.73/−0.71), and ineptitude with abuse (*r* = −0.13/−0.24). Among both men and women (Table 1), the ineptitude concern was the strongest, followed by the subjugation and abuse concerns (*M* (*SD*), men/women): 0.28 (0.76)/0.49 (0.87), 0.22 (0.48)/0.10 (0.58), −0.27 (0.40)/−0.32 (0.47). Table A1 in Appendix A presents the list of RFI items, as well as Cronbach’s alphas and centered means and standard deviations of the ten relationship fears for men and women.

**Confirmatory Factor Analysis.** The three-factorial model, including the subjugation, ineptitude, and abuse concerns as latent factors and ten relationship fears as observed variables, demonstrated a good fit: *χ*^2^ = 196, *df* = 30, *p* < *0*.001; *RMSEA* = 0.072 [90 % CI = 0.062–0.081]; *CFI* = 0.973; *TLI* = 0.959; *SRMR* = 0.032. All standardized estimates measuring connections between the latent and observed variables were above 0.59 (Table A2 in Appendix A). The three latent factors were negatively correlated with each other: subjugation with ineptitude (*r* = −0.38), subjugation with abuse (*r* = −0.70), and ineptitude with abuse (*r* = −0.19). Thus, the CFA corroborated the structure of the Relationship Fears Inventory (RFI) revealed by the MDS analysis.

### 3.2. The Connections Between Socio-Demographic Variables, Values, and Relationship Concerns

To test the hypothesized connections between socio-demographic variables, values, and relationship fears, we conducted path analysis using Mplus. Figure 2 presents the hypothesized model that included the following variables: socio-demographic variables (age, gender, education, religiosity, and ethnicity), two values axes (openness to change vs. conservation and self-transcendence vs. self-enhancement), and three relationship concerns (ineptitude, subjugation, and abuse). After the initial model was tested, aiming for the most parsimonious model, it was “trimmed”, i.e., all non-significant paths were excluded from the model.. The direct and indirect effects were tested using the bootstrapping method with 1000 re-samples with a 95% confidence interval.

The trimmed model demonstrated an excellent fit: *χ*^2^(14) = 12.6, *p* = 0.556; *RMSEA* (*CI*) = 0.000 (0.000; 0.027); *CFI* = 1.000; *TLI* = 1.002, SRMR = 0.012. The proportion of variance explained was significant for all three concerns: the ineptitude concern (8%), the subjugation concern (10%), and the abuse concern (13%). Figure 3 presents connections between variables (standardized estimates) in the trimmed model.

A higher preference for self-transcendence vs. self-enhancement values was connected to ineptitude (*β* = 0.17) and subjugation concerns (*β* = −0.08). A higher preference for openness to change vs. conservation values was connected to subjugation (*β* = 0.21) and abuse concerns (*β* = −0.23).

Age was directly connected to ineptitude (*β* = −0.11), subjugation (*β* = 0.18), and abuse concerns (*β* = −0.13). In addition, it was connected to openness to change vs. conservation (*β* = 0.08) and self-transcendence vs. self-enhancement values (*β* = 0.08). Age was indirectly connected to the ineptitude concern through self-transcendence vs. self-enhancement values (*β* = 0.014, *p* = 0.016) and to the abuse concern through openness to change vs. conservation values (*β* = −0.018, *p* = 0.017). The indirect effect of age on the subjugation concern was non-significant (*β* = 0.009, *p* = 0.205) because of its two indirect specific effects, through openness to change vs. conservation (*β* = 0.016, *p* = 0.018) and self-transcendence vs. self-enhancement values (*β* = −0.006, *p* = 0.031), compensated each other. The total effects of age on all three relationship concerns were significant: ineptitude (*β* = −0.095, *p* = 0.002), subjugation (*β* = 0.193, *p* < 0.001), and abuse (*β* = −0.148, *p* < 0.001).

Gender (1—*male*, 2—*female*) was directly connected to ineptitude (*β* = 0.11), subjugation (*β* = −0.08), and abuse concerns (*β* = −0.08). In addition, it was connected to self-transcendence vs. self-enhancement values (*β* = 0.11). Gender was indirectly connected to ineptitude (*β* = 0.018, *p* = 0.002) and subjugation concerns (*β* = −0.008, *p* = 0.009) through self-transcendence vs. self-enhancement values. The total effects of gender on all three concerns were significant: ineptitude (β = 0.124, *p* < 0.001), subjugation (*β* = −0.086, *p* = 0.003), and abuse (β = −0.081, *p* = 0.004).

Religiosity was not directly connected to any relationship concern. However, religiosity was connected to openness to change vs. conservation values (*β* = −0.24). Through these values, it was indirectly connected to subjugation (β = −0.049, *p* < 0.001) and abuse concerns (β = 0.055, *p* < 0.001).

Finally, ethnicity (1—*Palestinian*, 2—*Jewish*) was directly connected to ineptitude (*β* = 0.17) and abuse concerns (*β* = −0.14). In addition, it was connected to openness to change vs. conservation (*β* = 0.12) and self-transcendence vs. self-enhancement values (*β* = 0.26). Ethnicity was indirectly connected to ineptitude concern through self-transcendence vs. self-enhancement values (*β* = 0.044, *p* < 0.001) and to the abuse concern through openness to change vs. conservation values (*β* = −0.027, *p* = 0.001). The indirect connection between ethnicity and the subjugation concern was non-significant (*β* = 0.004, *p* = 0.699) because its two specific indirect effects, through openness to change vs. conservation (*β* = 0.024, *p* = 0.001) and self-transcendence vs. self-enhancement values (*β* = −0.020, *p* = 0.001), compensated each other. The total effects of ethnicity on ineptitude (*β* = 0.215, *p* < 0.001) and abuse (*β* = 0.170, *p* < 0.001) concerns were significant.

## 4. Discussion

In this study, we investigated the fear of romantic relationships. Based on Schwartz’s theory of values, we built a comprehensive inventory of fears young people considering romantic relationships may experience. We tested the fears’ structure and investigated their connections with personal values and socio-demographic variables.

### 4.1. The Relationship Fears’ Structure

The study results demonstrated that ten basic relationship fears might be combined into three clusters (relationship concerns). The ineptitude concern combines two fears: failing partner’s expectations and failing expectations of one’s relatives and friends. The subjugation concern combines four fears: loss of independence, boredom, sexual frustration, and thwarting one’s achievements. Finally, the abuse concern combines four fears: losing control over one’s resources, being hurt physically or sexually, harming relationships with one’s relatives and friends, and being accused of inappropriate behavior. The discovery of the three groups permitted us to reduce a large number of relationship fears to a limited number of concerns and use them in further analyses.

The ranking of concerns was identical among men and women, with ineptitude being the strongest concern, followed by subjugation and abuse. These findings indicate that the main reason for avoiding romantic relationships among men and women is their concern about not being good in relationships and not standing up to the expectations of close others. This finding indicates the need to teach young men and women relationship skills and strengthen their self-assertiveness in romantic relationships. For now, schools and mass media focus on physical and sexual violence and health hazards in romantic relationships, which are parts of the abuse concern ([13]). The present study results indicate that educators must address the ineptitude concerns of young people more.

Comparing relationship concerns between the two genders, we found that women had a stronger ineptitude concern, and men had stronger subjugation and abuse concerns. Women’s stronger ineptitude concern and men’s stronger subjugation concern have been reported in previous studies, reflecting the differences in social norms and expectations between men and women ([12]). However, the stronger abuse concern of men was unexpected. A possible explanation is related to the fact that previous studies focused on the fears of physical and sexual abuse and found them stronger among women ([6]; [14]); similar results have been obtained in the present study. However, in the present study, the abuse concern included fears of losing control over one’s resources and being accused of inappropriate behavior, and these fears were stronger among men. These fears among men may be related to the recent developments in protecting women’s rights, the MeToo movement, and changes in juridical practice ([17]; [28]). The present findings indicating that men and women may emphasize different aspects of abuse in romantic relationships are important for developing interventions that will be more tuned to the specific gender needs.

### 4.2. Relationship Concerns and Values

In this study, we revealed a meaningful pattern of connections between relationship concerns and personal values, which allowed us to understand the motivational foundations of relationship fears. Stronger ineptitude concern was associated with a higher preference for self-transcendence vs. self-enhancement values. This means that people caring for others are more likely to avoid romantic relationships because they fear not being good in them and disappointing their partner or significant others. Thus, they avoid relationships because they fear hurting others’ feelings. Such people may be oversensitive to the needs of others and have low self-esteem and a sense of mastery. Therefore, they may benefit from assertiveness training ([24]).

Stronger subjugation concern was associated with higher preferences for openness to change vs. conservation values. This means that people who highly value independence and autonomy and seek pleasure and variety in life avoid romantic relationships because they fear that relationships will prevent them from attaining these motivational goals. They perceive romantic relationships as restrictive and limiting one’s freedom. In addition, the subjugation concern was associated with a higher preference for self-enhancement vs. self-transcendence values. This means that people who prefer the advancement of their interests over the interests of others may sense that romantic relationships may subjugate them to others and harm the achievement of their goals. People with strong subjugation concerns may benefit from learning about the relationship motivations of the opposite gender in general and their potential partner in particular ([25], [26]). This knowledge may dismiss their concerns and better prepare them for relationships.

The abuse concern was associated with a higher preference for conservation vs. openness to change values. Thus, people who value security and stability may fear romantic relationships because they perceive them as potentially dangerous and threatening their physical and psychological well-being. According to the theory of values, conservation values are anxiety-driven ([21]). Therefore, people with strong abuse concerns might have had a history of abusive relationships in the past, which had increased their anxiety. Our results indicate that the abuse concerns are relatively low among young people. However, those with such concerns may have serious difficulties establishing romantic relationships and require professional help to overcome their problems.

### 4.3. Relationship Concerns and Socio-Demographic Variables

In the present study, we replicated some of the findings of previous studies regarding direct connections between socio-demographic variables and relationship concerns. In addition, we demonstrated that personal values partly mediate these connections.

***Age***. Age was directly connected to stronger subjugation concerns and weaker abuse and ineptitude concerns. Older people may be used to living without a romantic partner and may feel comfortable with this life. They may fear losing it in romantic relationships, which may explain their stronger subjugation concerns. In addition, older people may have more experience in romantic relationships and, therefore, develop a higher sense of mastery and self-esteem ([18]). This may explain their weaker abuse and ineptitude concerns. In addition, we found indirect effects of age on relationship concerns through values. Age was associated with higher preferences for openness to change vs. conservation and self-transcendence vs. self-enhancement and, through them, with stronger ineptitude and weaker abuse concerns. Two aspects of these findings are interesting. First, age is usually associated with a higher preference for conservation, not openness to change ([23]). The opposite connection found in our study may be explained by self-selection when people who have no romantic relationships at an advanced age are those with a higher preference for openness to change values. Consequently, such people are afraid of subjugation in romantic relationships. In addition, with age, people care more for others and, therefore, are more afraid of not being good in relationships and hurting others’ feelings. Importantly, in the present study, the direct and indirect effects of age on the ineptitude concern contradicted each other; however, the total effect of age on this concern was negative, i.e., the direct (negative) effect of age was stronger than its indirect (positive) effect.

**Gender.** Men reported stronger subjugation and abuse concerns, and women reported stronger ineptitude concerns. This pattern of connections between gender and relationship concerns corroborates the results of previous studies and reflects social norms and gender socialization, which stress men’s independence and strength and women’s weakness and dependency ([2]; [12]). In addition, similarly to previous studies, men reported a higher preference for self-enhancement vs. self-transcendence values ([23]). Through values, gender was indirectly connected to relationship concerns, so being a man was associated with stronger subjugation and being a woman—with a stronger ineptitude concern. These findings indicate that men perceive romantic relationships as more threatening to their general motivational goals related to self-advancement and power, while women perceive romantic relationships as more threatening to their goals of caring for others, strengthening their fear of being inadequate in attaining these goals.

***Religiosity***. Religiosity was not directly connected to relationship concerns. However, through its connection to a higher preference for conservation vs. openness to change values, religiosity was indirectly connected to subjugation (negatively) and abuse concerns (positively). All religions consider family a normative institution for adults and romantic relationships—a necessary step in creating a family ([29]). Moreover, religion assumes that family life imposes obligations and limits the freedom of both spouses ([9]). A higher preference for conservation values associated with higher religiosity may lead to more readiness to accept the restrictions relationships impose on personal freedom, which decreases subjugation concerns. On the other hand, the religiosity’s strengthening of the abuse concern may be related to the gender inequality in the hierarchy of roles and mutual obligations that are associated with the religious view of romantic relationships and the family ([15]).

***Ethnicity***. Palestinian Israelis reported stronger abuse and weaker ineptitude concerns compared with Jewish Israelis. The two groups did not differ in the subjugation concern. For several reasons, which include cultural, historical, and economic, Palestinian society is characterized by a higher level of violence compared to Jewish Israeli society ([4]; [5]). This may explain the stronger abuse concern in romantic relationships. At the same time, Palestinian society is more traditional than Jewish Israeli society; thus, it has more clear-cut norms of behavior related to the relations between genders ([16]). This may ensure a higher level of certainty in romantic relationships and thus decrease the ineptitude concern.

We found that ethnicity indirectly affected relationship concerns through values. Being Palestinian was associated with a stronger abuse concern through a higher preference for conservation values, and it was associated with a weaker ineptitude concern through a higher preference for self-enhancement values. As in previous studies ([25], [26]), we found that Palestinian Israelis reported higher preferences for conservation vs. openness to change and self-enhancement vs. self-transcendence values than Jewish Israelis. Thus, the differences between Palestinian and Jewish Israelis in relationship concerns may be partly explained by the value differences between the two groups.

### 4.4. Limitations and Suggestions for Further Research

Several limitations of the study must be considered. First, it was cross-sectional. Future longitudinal research would represent a significant advancement in the current findings. The second limitation of the study is its sample, which was not random. The lack of control over the sample may raise generalizability issues. Further research should be based on representative samples. The third limitation relates to the research population. The relationship fears inventory was based on interviews conducted in Israel, and our theoretical model connecting socio-demographic variables, values, and relationship concerns was tested only in one country. Testing it in other countries would be essential to its generalization. The fourth limitation of the study is its focus on individual-level factors and not investigating the macro and mezzo-level factors (e.g., friends and cultural norms) that might affect relationship fears. Fifth, we revealed the effect of values and socio-demographic variables on relationship fears. Further studies may investigate the effects of other variables, such as the relationships with parents, attachment style, history of previous relationships, sexuality development, interpersonal difficulties, self-awareness, and overprotection and avoidance traits, which may influence relationship fear. Finally, the present study focused on young people with no romantic partner. Further studies may investigate changes in relationship fears in other stages of romantic relationships: in the beginning, during cohabitation, and after marriage.

## 5. Conclusions

In this study, we investigated fears of romantic relationships. We revealed the existence of three clusters of relationship fears, which permitted us to group many relationship fears into a limited number of concerns and use them in further analysis. We confirmed that relationship concerns form a meaningful pattern of connections with personal values and demonstrated that people derive their relationship concerns from their general motivational goals expressed in values. Thus, the present study advances the theory of values connecting context-specific and general motivations. Our findings may be helpful in youth counseling to promote satisfactory decisions regarding romantic relationships. They may allow professionals to develop interventions facilitating the psychological adjustment of young people in the context of romantic relationships.

## Figures and Tables

**Figure 1 behavsci-15-00191-f001:**
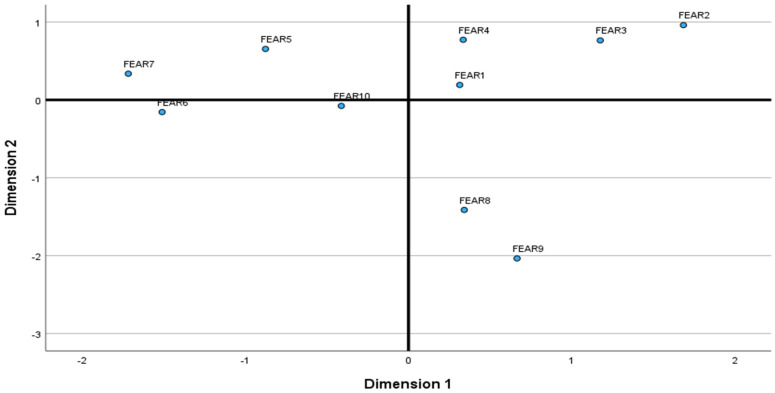
Multidimensional Scaling Configuration Derived in Two Dimensions. Note: Fear 1: To lose independence. Fear 2: To be bored. Fear 3: To be sexually frustrated. Fear 4: To thwart my achievements. Fear 5: To lose control over my resources. Fear 6: To be endangered physically and sexually. Fear 7: To be accused of inappropriate behavior. Fear 8: To fail my relatives’ and friends’ expectations. Fear 9: To fail my partner’s expectations. Fear 10: To harm relationships with my relatives and friends.

**Figure 2 behavsci-15-00191-f002:**
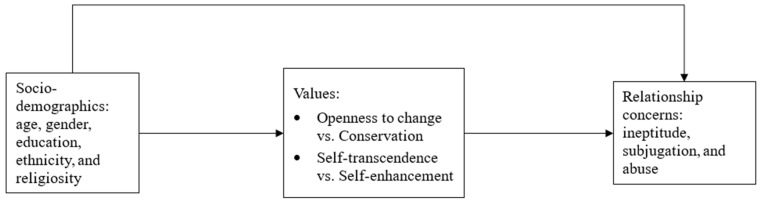
Theoretical Model.

**Figure 3 behavsci-15-00191-f003:**
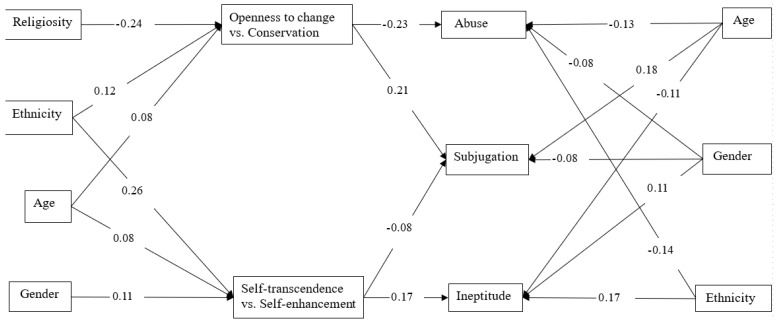
Path Analysis (Trimmed Model): Socio-Demographic Variables, Value Axes Scores, and Relationship Concerns (Standardized Estimates).

**Table 1 behavsci-15-00191-t001:** Correlations, Means, and Standard Deviations.

Variables	Subjugation	Ineptitude	Abuse	O2CHCONS	SETRSENH	Age	Gender	Education	Ethnicity	Religiosity
Subjugation	1									
Ineptitude	−0.376 ***	1								
Abuse	−0.705 ***	−0.202 ***	1							
O2CHCONS	0.245 ***	−0.007	−0.277 ***	1						
SETRSENH	−0.029	0.213 ***	−0.101 ***	0.091 **	1					
Age	0.213 ***	−0.040	−0.205 ***	0.162 ***	0.152 ***	1				
Gender	−0.106 ***	0.125 ***	−0.058	−0.016	0.089 **	−0.103 ***	1			
Education	0.044	−0.047	−0.033	−0.040	−0.047	0.284 ***	0.037	1		
Ethnicity	0.121 ***	0.167 ***	−0.258 ***	0.231 ***	0.280 ***	0.318 ***	−0.041	−0.179 ***	1	
Religiosity	−0.112 ***	−0.077	0.171 ***	−0.298 ***	−0.097 **	−0.209 ***	0.112 ***	0.060	−0.381 ***	1
*M(SD)*	0.15 (0.54)	0.40 (0.83)	−0.30 (0.44)	0.56 (.84)	0.83 (0.91)	25.1 (3.32)	1.57 (0.50)	13.6 (2.00)	0.72 (0.45)	2.32 (0.86)

Note: O2CHCONS—Openness to change vs. Conservation. SETRSENH—Self-Transcendence vs. Self-Enhancement. Gender: 1—men, 2—women. Ethnicity: 1—Palestinian Israelis, 2—Jewish Israelis. ** *p* < 0.01, *** *p* < 0.001.

## Data Availability

The data is available from the author.

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
