# Peer review of "Who Is Afraid of Romantic Relationships? Relationship Fears and Their Connection with Personal Values and Socio-Demographic Variables"

_behavsci, 2025, doi:10.3390/bs15020191_

Round 1
Reviewer 1 Report
Comments and Suggestions for Authors
The aim of the current study is to look for the structure and relevant factors associated with romantic relationship fear. The author has tested their complicated hypotheses in 1083 individuals. However, the author might also note the following comments on the study.
1. The general design can be simplified, e.g., a trial of romantic relationship fear in some participants; then, the validated questionnaire (with clear structures and names); then, a trial of the questionnaire in more participants, to seek the group (gender, age, background) differences, and to figure out the possible confounding factors which are prominent in different subscales of romantic relationship fear, in various age, or gender (sex); and then, the general conclusions to support the hypotheses.
2. Confounding factors can be understanding love (relationship), self-awareness or (over)protection, and avoidance traits, which will influence romantic relationship fear.
3. Where are the fear-measuring items? Are they exhaustive? The cluster analyses of the answers are not very ideal (according to Fig 1). Factor analysis might help, for instance, the latent factors and the questionnaire model forming, could offer the validated structure of the romantic relationship fear measuring. Item-loadings on latent factors can also provide more information.
4. What are the raw scores of the potential (different) groups, for instance, the gender (sex) differences? Or the age subgroups, or religious classifications?
5. Both section 3.4., and Fig 2, should be significantly reduced and simplified. Some hypotheses can be merged into one.
6. What will the results of the current study tell the reader? To offer information for the love cognition and education? The author might thus connect the current study to a love (romantic) theory.
Author Response
Reviewer 1:
- The general design can be simplified, e.g., a trial of romantic relationship fear in some participants; then, the validated questionnaire (with clear structures and names); then, a trial of the questionnaire in more participants, to seek the group (gender, age, background) differences, and to figure out the possible confounding factors which are prominent in different subscales of romantic relationship fear, in various age, or gender (sex); and then, the general conclusions to support the hypotheses.
We conducted the study in two stages. In stage one, we interviewed about 50 participants and gathered items measuring relationship fears. In the second stage, we tested the created Relationship Fears Inventory using MDS and CFA. According to modern literature, conducting exploratory and confirmatory tests in the same sample is legitimate (e.g., Schwartz, S. H., Cieciuch, J., Vecchione, M., Davidov, E., Fischer, R., Beierlein, C., ... & Konty, M. (2012). Refining the theory of basic individual values. Journal of Personality and Social Psychology, 103(4), 663. Wright, D., Treyvaud, K., Williams, L. A., & Giallo, R. (2022). Validation of the Karitane parenting confidence scale in measuring parental self-efficacy of Australian fathers. Journal of Child and Family Studies, 31(6), 1698-1706.). Large samples are preferable over small ones for statistical tests because reducing the sample size decreases the tests’ power. Thus, there is no reason to divide the large sample into two smaller ones. We explain the study design in the Introduction (p. 3): “The study consisted of three stages. First, we created a relationship fears inventory using scientific literature and personal interviews. After that, we tested the inventory’s structure using exploratory and confirmatory analyses. Finally, we investigated the connections between relationship fears, personal value preferences, and socio-demographic variables. We conducted a study in Israel using a large community sample of young Jews and Palestinians who had no romantic partners.”
- Confounding factors can be understanding love (relationship), self-awareness or (over)protection, and avoidance traits, which will influence romantic relationship fear.
There are a lot of factors affecting romantic fears. Some of them you mentioned, but there are also many others. In the present study, we focused on values and socio-demographic variables. However, following your comment, we included the following sentence in the Limitations and Suggestions for Further Research section (p. 22): “Fifth, we revealed the effect of values and socio-demographic variables on relationship fears. Further studies may investigate the effects of other variables, such as the relationships with parents, attachment style, history of previous relationships, sexuality development, interpersonal difficulties, self-awareness, and overprotection and avoidance traits, which may influence relationship fear.”
- Where are the fear-measuring items? Are they exhaustive? The cluster analyses of the answers are not very ideal (according to Fig 1). Factor analysis might help, for instance, the latent factors and the questionnaire model forming, could offer the validated structure of the romantic relationship fear measuring. Item-loadings on latent factors can also provide more information.
Table A1 in the Appendix presents the items measuring relationship fears. We did our best to create an exhaustive list of relationship fears, conducting a thorough literature search and interviewing dozens of young adults.
Following your suggestion, we conducted CFA of the three-factor model of relationship fears. Table A2 in the Appendix presents the connections between ten basic fears (observed variables) and the three relationship concerns (latent factors). The goodness of fit indexes of CFA are provided in the Results section (p. 13): ’Confirmatory Factor Analysis. The three-factorial model, including the subjugation, ineptitude, and abuse concerns as latent factors and ten relationship fears as observed variables, demonstrated a good fit: χ2 = 196, df = 30, p < .001; RMSEA = .072 [90 % CI = .062 - .081]; CFI = .973; TLI = .959; SRMR = .032. All standardized estimates measuring connections between the latent and observed variables were above .59 (Table A2 in the Appendix). The three latent factors were negatively correlated with each other: subjugation with ineptitude (r = -.38), subjugation with abuse (r = -.70), and ineptitude with abuse (r = -.19). Thus, the CFA corroborated the structure of the Relationship Fears Inventory (RFI) revealed by the MDS analysis.” The CFA has fully corroborated the three-factor structure of relationship fears obtained in the MDS analysis.
- What are the raw scores of the potential (different) groups, for instance, the gender (sex) differences? Or the age subgroups, or religious classifications?
We cannot provide group scores for every socio-demographic variable. However, following your comment, we provided the table of correlations, means, and standard deviations of all variables included in the study (Table 1).
- Both section 3.4., and Fig 2, should be significantly reduced and simplified. Some hypotheses can be merged into one.
We reduced and simplified the hypotheses (pp. 7-9):
- The structure of relationship fears. Relationship fears form a circumplex paralleling the values’ circumplex. Similarly to values, the fears’ circumplex may be partitioned into four clusters (relationship concerns), each associated with a higher-order value:
- The first cluster, called the subjugation concern, includes the fears of losing independence, being bored, and being sexually frustrated.
- The second cluster, called the depletion concern, includes the fears of thwarting one’s achievements and losing control over one’s resources.
- The third cluster, called the abuse concern, includes the following relationship fears: being endangered physically and sexually, being accused of inappropriate behavior, and harming relationships with relatives and friends.
- The fourth cluster, called the ineptitude concern, includes fears of failing a partner’s expectations and failing to meet the expectations of one’s relatives and friends.
- Relationship concerns derived from opposing poles of values axes (the subjugation and abuse and the depletion and ineptitude concerns) contradict each other and are located on opposite sides of the circumplex.
- Connections between values and relationship concerns:
- The subjugation concern is associated with a higher preference for openness to change vs. conservation values, and the abuse concern is associated with a higher preference for conservation vs. openness to change values.
- The depletion concern is associated with a higher preference for self-enhancement vs. self-transcendence values, and the ineptitude concern is associated with a higher preference for self-transcendence vs. self-enhancement values.
- Connections between socio-demographic variables and relationship concerns:
- Younger people have stronger inaptitude and abuse concerns.
- Men have stronger subjugation and depletion concerns, and women have stronger abuse concerns.
- Connections between socio-demographic variables and values:
- Being a man is associated with higher preferences for openness to change vs. conservation and self-enhancement vs. self-transcendence values.
- Age is associated with a higher preference for conservation vs. openness to change values.
- Education is associated with a higher preference for openness to change vs. conservation values.
- Religiosity is associated with a higher preference for conservation vs. openness to change values.
- Compared to Jewish Israelis, Palestinian Israelis have a higher preference for conservation vs. openness to change and self-enhancement vs. self-transcendence values.
- What will the results of the current study tell the reader? To offer information for the love cognition and education? The author might thus connect the current study to a love (romantic) theory.
The present study is not connected to the theory of love. Love usually develops at the beginning of relationships or when they are established. In the present study, we investigate fears that prevent establishing romantic relationships; this is a pre-relationships situation. The present study gives us a better understanding of aversive motivations in the context of romantic relationships. At a higher theoretical level, the study results allow us to understand how content-specific fears relate to general motivational goals. We added this explanation to the Abstract and Conclusion sections (p. 2): “The study results permit us to understand motivations to avoid romantic relationships. Thus, the present study advances the theory of values connecting context-specific and general motivations. The results obtained will be helpful in youth counseling to promote satisfactory decisions regarding romantic relationships.”
Reviewer 2 Report
Comments and Suggestions for Authors
Thank you for the opportunity to review the manuscript entitled “Who Is Afraid of Romantic Relationships? Relationship Fears, Personal Values, and Socio-Demographic Variables”. This study presents an interesting exploration of the structure of relationship fears and their associations with personal values and socio-demographic variables. The study’s strengths include the large sample size and the use of interviews to develop a valid measure for relationship concerns. However, I have some conceptual and methodological concerns that I believe, if addressed, could enhance the clarity and overall impact of the study.
1) The introduction and literature review provide relevant evidence from existing literature, which is helpful. However, the current structure feels more like a collection of isolated points rather than a cohesive narrative. It would be beneficial to synthesize the evidence more effectively and create a clear story line that guides readers through the rationale, innovation, and significance of the study. For example, the subsection titled “Theory of Human Values” is under “Group Differences in Relationship Fears,” but it primarily focuses on the theory itself. A more organized approach would enhance clarity and impact.
2) The introduction and lit review suggests a theory-driven approach using Personal Value Theory to guide the clustering of relationship fears, and to align the four higher-order values with four fear clusters. However, the analysis is more of a data-driven approach and yielded three clusters, which do not align with these theoretical categorizations. Could the authors clarify their conceptualization of relationship fears and values? For instance, are fears and values considered same constructs in different domains? If so, I think it is less meaningful to test their associations and indirect effects as they are the same constructs in nature. If not, could the authors elaborate the distinctions between these constructs and provide the rationale for applying value theory to cluster relationship fears?
3) The manuscript uses various terms, e.g., “Theory of Human Values,” “Values Theory,” “General Motivational Theory”, which seem refer to the same theory. I think use term consistently would improve clarity and readability.
4) The detailed introduction to the theory is helpful, but I have a question regarding the orthogonality of the axes in the value framework. I scanned Schwartz’s theory papers (2012, 2017), the axes of openness to change vs. conservation and self-enhancement vs. self-transcendence seem not orthogonal to me. Could the authors help me understand why these axes are presented as orthogonal in this study?
5) The detailed hypotheses are appreciated, but the section combines hypotheses with literature review. I think reorganize this section to separate hypotheses from the literature discussion could enhance clarity.
6) The examination of group differences in personal values and relationship fears is an important topic. Could the authors provide a rationale for the specific socio-demographic variables selected? Additionally, have the authors considered the intersectionality of socio-demographic variables? I think explore the interactions between groups could provide valuable insights.
7) It would be helpful to include more details on how socio-demographic variables are measured and coded in the method section.
Author Response
Reviewer 2:
1) The introduction and literature review provide relevant evidence from existing literature, which is helpful. However, the current structure feels more like a collection of isolated points rather than a cohesive narrative. It would be beneficial to synthesize the evidence more effectively and create a clear story line that guides readers through the rationale, innovation, and significance of the study. For example, the subsection titled “Theory of Human Values” is under “Group Differences in Relationship Fears,” but it primarily focuses on the theory itself. A more organized approach would enhance clarity and impact.
We reorganized the Introduction and literature review to better explain the study's rationale, innovation, and significance. We moved the Theory of Values section after the Previous Studies on Relationship Fears section.
2) The introduction and lit review suggests a theory-driven approach using Personal Value Theory to guide the clustering of relationship fears, and to align the four higher-order values with four fear clusters. However, the analysis is more of a data-driven approach and yielded three clusters, which do not align with these theoretical categorizations. Could the authors clarify their conceptualization of relationship fears and values? For instance, are fears and values considered same constructs in different domains? If so, I think it is less meaningful to test their associations and indirect effects as they are the same constructs in nature. If not, could the authors elaborate the distinctions between these constructs and provide the rationale for applying value theory to cluster relationship fears?
The fears and values are different-level constructs. Values refer to general motivational goals, which are relatively stable across time and situations, while relationship fears are context-specific. Following your comment, we stressed this distinction in the Introduction (p. 7): “Following Schwartz’s theory of values, we assumed that relationship fears express context-specific motivations. We further assumed that people avoid romantic relationships because they fear that such relationships may prevent them from attaining the general motivational goals expressed in their value preferences. Thus, we connected each relationship fear to a specific value, assuming that people with a high preference for a specific value experience a high level of relationship fear associated with this value. Finally, we assumed that the group differences in value preferences may explain group differences in relationship fears.”
3) The manuscript uses various terms, e.g., “Theory of Human Values,” “Values Theory,” “General Motivational Theory”, which seem refer to the same theory. I think use term consistently would improve clarity and readability.
Thank you for your comment. We now use the term “Theory of values” consistently.
4) The detailed introduction to the theory is helpful, but I have a question regarding the orthogonality of the axes in the value framework. I scanned Schwartz’s theory papers (2012, 2017), the axes of openness to change vs. conservation and self-enhancement vs. self-transcendence seem not orthogonal to me. Could the authors help me understand why these axes are presented as orthogonal in this study?
In all papers using the axes scores, the two axes are orthogonal. The same is true in the present study (see Table 1 in the revision). However, they do not have to, and an article by Rudnev et al. (2018) demonstrates that correlations between the higher-order values constituting the two axes vary cross-culturally. To be on the save side, we deleted the word orthogonal from the article.
5) The detailed hypotheses are appreciated, but the section combines hypotheses with literature review. I think reorganize this section to separate hypotheses from the literature discussion could enhance clarity.
We excluded the literature references from the hypotheses and simplified the hypotheses to enhance clarity (pp. 7-9):
- The structure of relationship fears. Relationship fears form a circumplex paralleling the values’ circumplex. Similarly to values, the fears’ circumplex may be partitioned into four clusters (relationship concerns), each associated with a higher-order value:
- The first cluster, called the subjugation concern, includes the fears of losing independence, being bored, and being sexually frustrated.
- The second cluster, called the depletion concern, includes the fears of thwarting one’s achievements and losing control over one’s resources.
- The third cluster, called the abuse concern, includes the following relationship fears: being endangered physically and sexually, being accused of inappropriate behavior, and harming relationships with relatives and friends.
- The fourth cluster, called the ineptitude concern, includes fears of failing a partner’s expectations and failing to meet the expectations of one’s relatives and friends.
- Relationship concerns derived from opposing poles of values axes (the subjugation and abuse and the depletion and ineptitude concerns) contradict each other and are located on opposite sides of the circumplex.
- Connections between values and relationship concerns:
- The subjugation concern is associated with a higher preference for openness to change vs. conservation values, and the abuse concern is associated with a higher preference for conservation vs. openness to change values.
- The depletion concern is associated with a higher preference for self-enhancement vs. self-transcendence values, and the ineptitude concern is associated with a higher preference for self-transcendence vs. self-enhancement values.
- Connections between socio-demographic variables and relationship concerns:
- Younger people have stronger inaptitude and abuse concerns.
- Men have stronger subjugation and depletion concerns, and women have stronger abuse concerns.
- Connections between socio-demographic variables and values:
- Being a man is associated with higher preferences for openness to change vs. conservation and self-enhancement vs. self-transcendence values.
- Age is associated with a higher preference for conservation vs. openness to change values.
- Education is associated with a higher preference for openness to change vs. conservation values.
- Religiosity is associated with a higher preference for conservation vs. openness to change values.
- Compared to Jewish Israelis, Palestinian Israelis have a higher preference for conservation vs. openness to change and self-enhancement vs. self-transcendence values.
6) The examination of group differences in personal values and relationship fears is an important topic. Could the authors provide a rationale for the specific socio-demographic variables selected? Additionally, have the authors considered the intersectionality of socio-demographic variables? I think explore the interactions between groups could provide valuable insights.
There was no specific rationale for selecting the socio-demographic variables in this research. We included those that have been used in previous studies on values. Exploring the effect of intersectionality on values is a great idea. It has not been done previously, to the best of our knowledge. However, this is an issue for a separate study.
7) It would be helpful to include more details on how socio-demographic variables are measured and coded in the method section.
We included a sub-section on socio-demographic variables in the Instruments section (p. 11): “Socio-demographic variables. Five socio-demographic variables were included in the analysis: age, gender (1 – man, 2 – woman), education (number of years studied), ethnicity (1 – Palestinian Israeli, 2 – Jewish Israeli), and religiosity (1 – atheist, 2 – secular, 3 – traditional, and 4 – religious).”
Reviewer 3 Report
Comments and Suggestions for Authors
I enjoyed the subject of this paper, and it has potential. However, it could be more precise and organised. I recommend the authors split it into two parts: one on instrument adaptation and validation and another on the correlation between variables. The paper should also be more organised in terms of writing, organisation, and scientific structure. I don’t recommend immediate publication, but I hope the author can make the changes and resubmit.
Title and Abstract
I recommend that the authors rewrite the title and abstract according to APA guidelines. They should also clarify the study's primary goal and the author’s creation of a psychological measure. The results should be resumed, and there should be a conclusion with implications.
Introduction
The introduction isn’t organised, and the information isn’t ordered. It doesn’t adhere to the guidelines for a well-structured introduction in a scientific article. For example, Point 1 of the introduction doesn’t make sense. Part of the information can be introduced in point 2 (the reference to the model is repeated), and the second part should be at the end of the theory section before the hypothesis. Point 3 repeats information from point 2. The point 3.3. should be presented in the procedures or measures part. There is a lot of pertinent information in this part of the paper. Still, it should be resumed, organised and integrated to explain more precisely the innovation and the state of the art related to the objectives of this study.
The authors present many hypotheses, which need to be clarified and made easier to understand. If the authors explain that the studies regarding relationship fears are not theory-based, how can so many hypotheses be explored? Also, how… Do they relate to the study objectives? This needs clarification. The literature presented doesn’t support all these hypotheses, namely in the culture of the population of this study. Also, is this a study or two studies in these hypotheses? It isn’t very clear.
Method
The sample includes adults aged 18 to 30, which may represent very different patterns of past relationships and related life episodes. Was this explored or asked? How do the authors control past experiences in relationships? How did this influence inclusion and exclusion factors?
Some information from procedures should be in the sample.
Did the authors use a sociodemographic questionnaire?
The authors didn’t present a description of the statistical analysis.
Results
The authors must clarify the decision to apply multidimensional scaling (MDS) to the questionnaire's structure. I understand the rationale, but it is incomplete, and this instrument deserves a more complex discussion on its validation. Did the authors consider alternative methods for this data type, such as Principal Component Analysis or Factor analysis? MDS doesn’t account for the strength of intercorrelations, which may be necessary regarding the validity of the measurement of these types of constructs. With MDS, it can be difficult to distinguish between closely related variables, and this may lead to the loss of meaningful variance in data. This may be a very interesting questionnaire and should be tested for its reliability and validity.
Discussion
This section is the best part of the paper because it is more organised than the rest. The discussion only gives a better sense of the whole paper. However, there are many results, and it’s hard to understand the main direction and implications of these results. There is also a lot of information, which is sometimes repeated. How does the author give a final sense of the fear and the values? What are the social and clinical implications of these results? What changes are in the theoretical framework of romantic relationships?
The result is men feeling more fear of abuse. I think that this result should be more explored since we have cultural changes in gender roles, for example. I believe that some results are very simplistically discussed. Several variables can be related and may mediate some of these results, such as the history of parents’ relationships, affect, attachment style, history of relationships, sexuality development, and interpersonal difficulties.
The authors should clarify the limitations regarding the questionnaire adaptation and validation. Also, the data doesn’t allow a linear conclusion, such as the findings indicating that people fear romantic relationships because they worry they may harm their attainment of important goals. I think the writing should be humbler and exploratory.
Author Response
Reviewer 3:
Title and Abstract
I recommend that the authors rewrite the title and abstract according to APA guidelines. They should also clarify the study's primary goal and the author’s creation of a psychological measure. The results should be resumed, and there should be a conclusion with implications.
Following your recommendation, we retitled the article: “Who Is Afraid of Romantic Relationships? Relationship Fears and Their Connection with Personal Values and Socio-Demographic Variables.”
We also rewrote the abstract, clarifying the study’s goals and the creation of a psychological measure: “This study investigates the fears of romantic relationships. Based on Schwartz’s theory of values, we built a comprehensive inventory of fears young people seeking romantic relationships may experience. We tested the fears’ structure and the connections with personal values and socio-demographic variables. … The present study advances the theory of values connecting context-specific and general motivations. The results obtained will be helpful in youth counseling to promote satisfactory decisions regarding romantic relationships.”
Finally, we summarized the results and added a conclusion with theoretical and practical implications (p. 23): “In this study, we investigated the fear of romantic relationships. We revealed the existence of three clusters of relationship fears, called ineptitude, subjugation, and abuse concerns. This permitted us to assemble many relationship fears into a limited number of concerns and use them in further analysis. We confirmed that relationship concerns form a meaningful pattern of connections with personal values and demonstrated that people derive their relationship concerns from their general motivational goals expressed in values. Thus, the present study advances the theory of values connecting context-specific and general motivations. Our findings may be helpful in youth counseling to promote satisfactory decisions regarding romantic relationships. They may allow professionals to develop interventions facilitating the psychological adjustment of young people in the context of romantic relationships.”
Introduction.
The introduction isn’t organised, and the information isn’t ordered. It doesn’t adhere to the guidelines for a well-structured introduction in a scientific article. For example, Point 1 of the introduction doesn’t make sense. Part of the information can be introduced in point 2 (the reference to the model is repeated), and the second part should be at the end of the theory section before the hypothesis. Point 3 repeats information from point 2. The point 3.3. should be presented in the procedures or measures part. There is a lot of pertinent information in this part of the paper. Still, it should be resumed, organised and integrated to explain more precisely the innovation and the state of the art related to the objectives of this study.
We rewrote the Introduction following the guidelines provided in your comment (p. 3): “Romantic relationships are interpersonal interactions based on emotional and physical attraction that could lead to a long-term intimate connection (August et al., 2023). Romantic relationships fulfill basic human needs in being loved, feeling safe, and bonding with a significant other; therefore, they are paramount for individual well-being (Adamczyk, 2022). However, many adolescents and young adults do not have and do not want a romantic partner, and a growing number of people worldwide remain single throughout their lives (Brown, 2020; Ortiz-Ospina, 2019; Tang, 2019). Factors averting people from romantic relationships have been scarcely investigated. In this study, we are trying to understand what people fear in romantic relationships.
The study aims to investigate the motivational foundations of relationship fears and, at a more abstract theoretical level, to reveal the connections between context-specific and general motivations. The study is based on the theory of values (Schwartz, 2017), which considers personal values as cognitive constructs reflecting general motivational goals and directing people’s cognition and behavior (Sagiv & Roccas, 2021). The study consisted of three stages. First, we created a relationship fears inventory using scientific literature and personal interviews. After that, we tested the inventory’s structure using exploratory and confirmatory analyses. Finally, we investigated the connections between relationship fears, personal value preferences, and socio-demographic variables. We conducted a study in Israel using a large community sample of young Jews and Palestinians who had no romantic partners.”
The authors present many hypotheses, which need to be clarified and made easier to understand. If the authors explain that the studies regarding relationship fears are not theory-based, how can so many hypotheses be explored? Also, how… Do they relate to the study objectives? This needs clarification. The literature presented doesn’t support all these hypotheses, namely in the culture of the population of this study. Also, is this a study or two studies in these hypotheses? It isn’t very clear.
We simplified the hypotheses and presented them in a more orderly way (pp. 7-9):
- The structure of relationship fears. Relationship fears form a circumplex paralleling the values’ circumplex. Similarly to values, the fears’ circumplex may be partitioned into four clusters (relationship concerns), each associated with a higher-order value:
- The first cluster, called the subjugation concern, includes the fears of losing independence, being bored, and being sexually frustrated.
- The second cluster, called the depletion concern, includes the fears of thwarting one’s achievements and losing control over one’s resources.
- The third cluster, called the abuse concern, includes the following relationship fears: being endangered physically and sexually, being accused of inappropriate behavior, and harming relationships with relatives and friends.
- The fourth cluster, called the ineptitude concern, includes fears of failing a partner’s expectations and failing to meet the expectations of one’s relatives and friends.
- Relationship concerns derived from opposing poles of values axes (the subjugation and abuse and the depletion and ineptitude concerns) contradict each other and are located on opposite sides of the circumplex.
- Connections between values and relationship concerns:
- The subjugation concern is associated with a higher preference for openness to change vs. conservation values, and the abuse concern is associated with a higher preference for conservation vs. openness to change values.
- The depletion concern is associated with a higher preference for self-enhancement vs. self-transcendence values, and the ineptitude concern is associated with a higher preference for self-transcendence vs. self-enhancement values.
- Connections between socio-demographic variables and relationship concerns:
- Younger people have stronger inaptitude and abuse concerns.
- Men have stronger subjugation and depletion concerns, and women have stronger abuse concerns.
- Connections between socio-demographic variables and values:
- Being a man is associated with higher preferences for openness to change vs. conservation and self-enhancement vs. self-transcendence values.
- Age is associated with a higher preference for conservation vs. openness to change values.
- Education is associated with a higher preference for openness to change vs. conservation values.
- Religiosity is associated with a higher preference for conservation vs. openness to change values.
- Compared to Jewish Israelis, Palestinian Israelis have a higher preference for conservation vs. openness to change and self-enhancement vs. self-transcendence values.
Method
The sample includes adults aged 18 to 30, which may represent very different patterns of past relationships and related life episodes. Was this explored or asked? How do the authors control past experiences in relationships? How did this influence inclusion and exclusion factors?
We did not collect information about previous relationship experiences. The inclusion criteria were age 18-30 (to focus on emerging adults, when stable romantic relationships are age-normative) and not having a romantic relationship in the present (to exclude the effect of the present relationships on relationship fears). The effect of previous relationships may be a topic for a separate study, which must include investigating the effects of their quantity and quality.
Some information from procedures should be in the sample.
We moved the sentence “Adults aged 18-30 who did not have a romantic partner were invited to participate in the study.” to the Sample section.
Did the authors use a sociodemographic questionnaire?
We did. We added the description of socio-demographic variables in the Instruments section (p. 11): “Socio-demographic variables. Five socio-demographic variables were included in the analysis: age, gender (1 – man, 2 – woman), education (number of years studied), ethnicity (1 – Palestinian Israeli, 2 – Jewish Israeli), and religiosity (1 – atheist, 2 – secular, 3 – traditional, and 4 – religious).”
The authors didn’t present a description of the statistical analysis.
We created a section describing the statistical analyses (pp. 11-12): “We tested the structure of relationship fears in two steps. First, we calculated the mean-centered scores for the ten relationship fears using the appropriate items and applied multidimensional scaling (MDS) to the ten fears. We used the Multidimensional Scale module in SPSS to conduct MDS. After establishing the structure in MDS, we retested it using a Confirmatory Factor Analysis (CFA) of the ten relationship fears (for a similar approach for testing circumplex models, see Cieciuch, 2017; Czyżkowska & Cieciuch, 2020; Schwartz et al., 2012). After that, we used Structural Equation Modeling (SEM) to test the connections between socio-demographic variables, values, and relationship fears. We conducted CFA and SEM using Mplus (Muthén & Muthén, 2012).”
Results
The authors must clarify the decision to apply multidimensional scaling (MDS) to the questionnaire's structure. I understand the rationale, but it is incomplete, and this instrument deserves a more complex discussion on its validation. Did the authors consider alternative methods for this data type, such as Principal Component Analysis or Factor analysis? MDS doesn’t account for the strength of intercorrelations, which may be necessary regarding the validity of the measurement of these types of constructs. With MDS, it can be difficult to distinguish between closely related variables, and this may lead to the loss of meaningful variance in data. This may be a very interesting questionnaire and should be tested for its reliability and validity.
Schwartz and his colleagues consider MDS more appropriate for testing circumplex scales than exploratory factor analysis (Schwartz et al., 2012). In addition, other researchers used this approach to test circumplex scales (Cieciuch, 2017; Czyżkowska & Cieciuch, 2020). Following your and other reviewers’ comments, we conducted a two-stage analysis, first MDS and, after that, CFA. The CFA fully corroborated the structure obtained by MDS. We described the CFA as follows (p. 13): “The three-factorial model, including the subjugation, ineptitude, and abuse concerns as latent factors and ten relationship fears as observed variables, demonstrated a good fit: χ2 = 196, df = 30, p < .001; RMSEA = .072 [90 % CI = .062 - .081]; CFI = .973; TLI = .959; SRMR = .032. The three latent factors were negatively correlated with each other: subjugation with ineptitude (r = -.38), subjugation with abuse (r = -.70), and ineptitude with abuse (r = -.19). All standardized estimates measuring connections between the latent and observed variables were above .59 (Table A2 in the Appendix). Thus, the CFA corroborated the structure of the Relationship Fears Inventory (RFI) revealed by the MDS analysis.”
Discussion
This section is the best part of the paper because it is more organised than the rest. The discussion only gives a better sense of the whole paper. However, there are many results, and it’s hard to understand the main direction and implications of these results. There is also a lot of information, which is sometimes repeated. How does the author give a final sense of the fear and the values? What are the social and clinical implications of these results? What changes are in the theoretical framework of romantic relationships?
We deleted repetitions and discussed the social and clinical implications of the results obtained. We also stressed our findings' theoretical and practical significance in the Conclusion section (p. 23): “In this study, we investigated the fear of romantic relationships. We revealed the existence of three clusters of relationship fears, called ineptitude, subjugation, and abuse concerns. This permitted us to group many relationship fears into a limited number of concerns and use them in further analysis. We confirmed that relationship concerns form a meaningful pattern of connections with personal values and demonstrated that people derive their relationship concerns from their general motivational goals expressed in values. Thus, the present study advances the theory of values connecting context-specific and general motivations. Our findings may be helpful in youth counseling to promote satisfactory decisions regarding romantic relationships. They may allow professionals to develop interventions facilitating the psychological adjustment of young people in the context of romantic relationships.”
The result is men feeling more fear of abuse. I think that this result should be more explored since we have cultural changes in gender roles, for example. I believe that some results are very simplistically discussed. Several variables can be related and may mediate some of these results, such as the history of parents’ relationships, affect, attachment style, history of relationships, sexuality development, and interpersonal difficulties.
We explored the results related to gender differences in relationship fears more deeply (p. 17): “Comparing relationship concerns between the two genders, we found that women had stronger ineptitude concerns, and men had stronger subjugation and abuse concerns. The women’s stronger ineptitude concerns and men’s stronger subjugation concerns have been reported in previous studies, reflecting different social norms and expectations from men and women (Glickman & La Greca, 2004). However, the stronger abuse concern of men was unexpected. A possible explanation is related to the fact that previous studies focused on the fears of physical and sexual abuse and found them stronger among women (Brooks et al., 2020; Leemis et al., 2022); similar results have been obtained in the present study. However, in the present study, the abuse concern included fears of losing control over one’s resources and being accused of inappropriate behavior, and these fears were stronger among men. These fears among men may be related to the recent developments in protecting women’s rights, the Metoo movement, and changes in juridical practice (Nutbeam & Mereish, 2022; Tippett, 2018). The present findings indicating that men and women may emphasize different aspects of abuse in romantic relationships are important for developing interventions that will be more tuned to the specific gender needs.”
Exploring the effect of other factors on relationship fears is beyond the scope of the present article. We included it as a limitation of the present study (p. 22): “Further studies may investigate the effects of other variables, such as the relationships with parents, attachment style, history of previous relationships, sexuality development, interpersonal difficulties, self-awareness, and overprotection and avoidance traits, which may influence relationship fear.”
The authors should clarify the limitations regarding the questionnaire adaptation and validation. Also, the data doesn’t allow a linear conclusion, such as the findings indicating that people fear romantic relationships because they worry they may harm their attainment of important goals. I think the writing should be humbler and exploratory.
We deleted the sentence about worries, as you suggested. We stressed the sample limitations regarding the fears inventory (p. 22): “The third limitation relates to the research population. The relationship fears inventory was based on interviews conducted in Israel, and our theoretical model connecting socio-demographic variables, values, and relationship fears was tested only in one country.”
Round 2
Reviewer 1 Report
Comments and Suggestions for Authors
No further comments